# The Lysosomal Sequestration of Tyrosine Kinase Inhibitors and Drug Resistance

**DOI:** 10.3390/biom9110675

**Published:** 2019-10-31

**Authors:** Eliska Ruzickova, Nikola Skoupa, Petr Dolezel, Dennis A. Smith, Petr Mlejnek

**Affiliations:** 1Department of Anatomy, Faculty of Medicine and Dentistry, Palacky University Olomouc, Hnevotinska 3, Olomouc 77515, Czech Republic; elis.ruzickova@gmail.com (E.R.); NikolaSkoupa@seznam.cz (N.S.); p.dolezel@atlas.cz (P.D.); 2Honorary Professor, Department of Chemistry, University of Capetown, Cape Town 7700, South Africa; dennissmith55009@gmail.com

**Keywords:** tyrosine kinase inhibitors, lysosomal sequestration, drug resistance, target sites, extralysosomal space, extracellular space

## Abstract

The Lysosomal sequestration of weak-base anticancer drugs is one putative mechanism for resistance to chemotherapy but it has never been directly proven. We addressed the question of whether the lysosomal sequestration of tyrosine kinase inhibitors (TKIs) itself contributes to the drug resistance in vitro. Our analysis indicates that lysosomal sequestration of an anticancer drug can significantly reduce the concentration at target sites, only when it simultaneously decreases its extracellular concentration due to equilibrium, since uncharged forms of weak-base drugs freely diffuse across cellular membranes. Even though the studied TKIs, including imatinib, nilotinib, and dasatinib, were extensively accumulated in the lysosomes of cancer cells, their sequestration was insufficient to substantially reduce the extracellular drug concentration. Lysosomal accumulation of TKIs also failed to affect the Bcr-Abl signaling. Cell pre-treatment with sunitinib significantly enhanced the lysosomal accumulation of the TKIs used; however, without apparent lysosomal biogenesis. Importantly, even increased lysosomal sequestration of TKIs neither decreased their extracellular concentrations nor affected the sensitivity of Bcr-Abl to TKIs. In conclusion, our results clearly show that the lysosomal sequestration of TKIs failed to change their concentrations at target sites, and thus, can hardly contribute to drug resistance in vitro.

## 1. Introduction

Some studies suggest that, in addition to ABC transporters ABCB1, ABCC1, and ABCG2, other mechanisms lead to decreased intracellular drug accumulation, such as lower extracellular and higher intracellular pH [1,2] or lysosomal exocytosis of anticancer drugs sequestered in lysosomes [3,4]. Decreased cytotoxicity due to the altered intracellular drug distribution associated with increased lysosomal sequestration [5,6] is another option.

Lysosomes have been considered mediators of anthracycline drug resistance for several decades [7,8,9,10,11]. Altered intracellular distribution of anthracyclines in cancer cells, associated with resistance, was usually verified using fluorescence microscopy [8,9,10,11]. However, different approaches have provided contradictory results. For example, Noel et al. demonstrated that nuclear accumulation of daunorubicin and doxorubicin is equal, despite different accumulations in lysosomes [12]. In addition, it was demonstrated that drug sequestration in cytoplasmatic organelles does not affect cell sensitivity to anthracyclines in multidrug resistant cells [13].

Later on, it was shown that lysosomal sequestration of several TKIs may also affect cell sensitivity to these drugs [14,15,16,17,18]. Since the intracellular distribution of TKIs cannot be easily monitored using fluorescence microscopy, several indirect methods are used. Importantly, published results do not unambiguously support the hypothesis that the lysosomal sequestration of TKIs itself can mediate resistance to these drugs. For example, lysosomal sequestration of sunitinib (SUN) did not affect pro-proliferative signaling mediated by p-ERK1/2 and p-Akt [15]. Similarly, no effect of inhibition of lysosomal sequestration of imatinib (IM) on c-Kit signaling was observed [19]. Such results are, however, in a sharp contrast to the postulation of Zhitomirsky and Assaraf that lysosomal sequestration of hydrophobic weak-base drugs prevents their accessibility to intracellular target sites, resulting in drug resistance [20].

Resistance to chemotherapeutic agents is a widespread problem in cancer treatment. Whilst the outcomes are of prime importance in vivo, investigation of the mechanism usually involves in vitro experiments. Reversible binding and sequestration are important and need to be taken into account in the static in vitro situation. However, there is a fundamental difference between the dynamic in vivo and in vitro situation. Reversible binding and sequestration are irrelevant under steady state, serial dosing conditions (which applies to almost all oncology pharmaceutical interventions) and cannot be mechanisms of resistance. In this situation, the extracellular free drug concentration is controlled by intrinsic clearance, which in turn controls the intracellular drug concentration, influenced also by the membrane permeability of the drug and the presence of transporter systems [21,22].

In this paper we specifically address the question of the quantities of TKIs, IM, NIL, and DAS, that are sequestered in lysosomes and whether the mechanism itself is sufficient to mediate significant resistance in cancer cells in vitro. Our approach was based on the following hypothesis. If lysosomal sequestration mediates drug resistance due to reduced concentration at target sites in vitro as suggested [5,6], at least two conditions must be fulfilled. First, according to our analysis demonstrated in Figure 1, lysosomal sequestration of an anticancer drug that reduces drug concentration at target sites (in extralysosomal space) in vitro must simultaneously decrease its extracellular level due to equilibrium, since uncharged forms of weak-base drugs freely diffuse across cellular membranes [5,20]. We assessed that by measuring lysosomal accumulation and comparing it to added drug concentration. Second, the lysosomal drug sequestration must significantly decrease the sensitivity of molecular target to the drug [5,20]. This was assessed by monitoring of Bcr-Abl signaling. To the best of our knowledge, the first condition was not met, although it is essential. Unfortunately, the second condition was never observed. In fact, we and others found that lysosomal sequestration does not change the sensitivity of a molecular target to its sequestrated drug [15,19].

Our results clearly indicate that within in vitro experiments, it is difficult to establish the causal relationship between lysosomal sequestration of TKIs, IM, NIL, and DAS, and drug resistance because the total accumulation capacity of lysosomes for the TKIs studies is too low to substantially reduce their extralysosomal concentrations, and decrease the sensitivity of target kinases to these drugs. In accordance with these findings, we observed that lysosomal sequestration of IM, NIL, and DAS had no effect on Bcr-Abl signaling in human leukemia cells.

## 2. Materials and Methods

### 2.1. Chemicals

Imatinib mesylate (STI571, Gleevec; IM, purity ≥ 98%) and nilotinib hydrochloride (Tasigna, AMN107, NIL) were kindly provided by Novartis (Basel, Switzerland). Dasatinib hydrochloride, (Sprycel, DAS; purity ≥ 98%) and sunitinib malate (sutent, SUN) were purchased from Selleckchem (Huston, TX, USA).

### 2.2. Cell Culture

Human chronic myelogenous leukemia K562 cells were cultured in the RPMI-1640 medium supplemented with a 10% calf fetal serum and antibiotics in 5% CO_2_ atmosphere at 37 °C. Human lung adenocarcinoma A549 cells and human breast adenocarcinoma MCF-7 cells were cultured in Dulbecco’s MEM modified supplemented with a 10% calf fetal serum and antibiotics in 5% CO_2_ atmosphere at 37 °C. All cell lines were obtained from the European Collection of Authenticated Cell Cultures (ECACC, Salisbury, UK).

### 2.3. Assay for Determination of Intracellular IM Levels

Cells (density of 4 × 10^5^/mL) were incubated in the growth medium with appropriate IM concentration in the absence or presence of BafA1 for 3 h in 5% CO_2_ atmosphere at 37 °C. Cell pellets were extracted using ice cold 4% (*w/v*) formic acid (FA) in water on their separation from the growth medium by centrifugation through a layer of silicone oil. Cell extracts were clarified by centrifugation (40,000 × *g* 10 min at 4 °C), diluted with extraction solution, and analyzed by liquid chromatography coupled with low-energy collision tandem mass spectrometry (LC/MS/MS). Details are given elsewhere [23,24].

### 2.4. Assay for Determination of Intracellular NIL Levels

Cells (density of 5 × 10^5^/mL) were incubated in the growth medium with appropriate NIL concentration in the absence or presence of BafA1 for 3 h in 5% CO_2_ atmosphere at 37 °C. Cell pellets were extracted using ice cold 1% (*w*/*v*) FA in 50% (*v*/*v*) methanol in water upon their separation from the growth medium by centrifugation through a layer of silicone oil. Cell extracts were clarified by centrifugation (40,000 × *g* 10 min at 4 °C), diluted with extraction solution, and analyzed by liquid chromatography coupled with low-energy collision tandem mass spectrometry (LC/MS/MS). Details are given elsewhere [25].

### 2.5. Assay for Determination of Intracellular DAS Levels

Cells (density of 5 × 10^5^/mL) were incubated in the growth medium with appropriate DAS concentration in the absence or presence of BafA1 for 3 h in a 5% CO_2_ atmosphere at 37 °C. Cell pellets were extracted using ice cold 1% (*w/v*) FA in 50% (*v/v*) methanol in water upon their separation from the growth medium by centrifugation through a layer of silicone oil. Cell extracts were clarified by centrifugation (40,000 × *g* 10 min at 4 °C), diluted with extraction solution, and analyzed by liquid chromatography coupled with low-energy collision tandem mass spectrometry (LC/MS/MS). Details are given elsewhere [26].

### 2.6. Calculation of TKIs in Lysosomes

Absolute accumulation of TKIs in lysosomes was calculated as follows. The value of the intracellular accumulation of a particular TKI in the presence of BafA1 (drug content in the cell except lysosomes), an inhibitor of vacuolar H(+)-ATPase [27], was subtracted from the value of intracellular accumulation of particular TKI in the absence of BafA1 (drug content in the cell including lysosomes) [19]. Absolute accumulation of TKI in lysosomes was expressed as the molar amount of a particular TKI in lysosomes per 10^6^ cells. Relative accumulation of TKIs was calculated as follows. The absolute value of the accumulation of a particular TKI in lysosomes was divided by the value of intracellular accumulation of particular TKI. Relative accumulations of TKIs are expressed in percentages.

### 2.7. Western Blot Analysis

Preparation and processing of protein samples were done as described elsewhere [28]. Briefly, cells were washed in ice cold phosphate buffered saline (PBS) and proteins were extracted using lysis buffer (50 mM Tris/HCl buffer pH 8.1 containing 1% NP-40, 150 mM NaCl, 50 mM NaF, 5 mM EDTA, and 5 mM sodium pyrophosphate, supplemented with protease (Roche, Mannheim, Germany) and phosphatase (Sigma-Aldrich, Saint Louis, MO, USA) inhibitor cocktails). Cell extracts were heat denatured in Laemmli buffer (31.25 mM Tris/HCl, pH = 6.8, 10% glycerol, 2% SDS, 5% 2-mercaptoethanol, 0.005% bromphemol blue). Samples (30 μg protein) were separated by SDS-PAGE on 10% gels and transferred onto nitrocellulose membranes.

Lysosomal proteins were analyzed using rabbit monoclonal anti-LAMP1 (D2D11) XP antibody (1:1000), rabbit monoclonal anti-LAMP2 (D5C2P) antibody (1:1000), and rabbit monoclonal anti-ATP6V1B2 (D2F9R) antibody ((1:1000) Cell Signaling Technology, Denvers, MA, USA).

Bcr-Abl signaling was analyzed using rabbit polyclonal anti-phospho-Bcr (Tyr177) antibody (1:1000) and rabbit polyclonal anti-phospho-CrkL (Tyr207) antibody ((1:1000); Cell Signaling Technology, Denvers, MA, USA).

To confirm equal protein loading, immunodetection was performed with the rabbit polyclonal anti-actin antibody ((1:2000) Sigma-Aldrich, St. Louis, MO, USA). The signal was detected using a horseradish peroxidase-conjugated secondary antibody (1:15,000) Dako, Glostrup, Denmark). Products were visualized using enhanced chemiluminescence (ECL; Amersham, Little Chalfont, UK).

### 2.8. Activity of Lysosomal Hydrolases

Cells were washed in ice cold PBS and lysed in buffer (25 mM HEPES (pH 7.0), 0.5% CHAPS, 0.5% Triton X-100, 0.5 mM dithiothreitol, 2 mM EGTA, and protease cocktail inhibitors) on ice for 30 min. Cell extracts were clarified by centrifugation (15,000 rpm/15 min/4 °C). Protein concentration was determined by the Bradford method [29]. The enzymatic reaction was initiated by adding cell extract (equivalent of 100 μg protein) to the cell assay buffer (50 mM NaCl, 50 mM citrate–phosphate buffer, pH 4.5) containing appropriate substrate. Acidic phosphatase (ACP) activity was measured using 1 mM 4-methylumbelliferyl phosphate as a substrate. β-D-galactosidase (GLB) activity was measured using 1 mM 4-methylumbelliferyl-β-D-galactopyranoside as a substrate. The reaction mixture was incubated at 37 °C for 30 min, and then the enzymatic reaction was stopped by adding 1 M Tris buffer, pH 10.7. The relative fluorescence of released 4-methylumbelliferone was monitored at 365/445 nm.

### 2.9. LysoTracker Red Staining

Cells were incubated with a medium containing 300 nM LysoTracker Red DND-99 (Invitrogen) for 30 min at 37 °C and then analyzed using flow cytometry (488/580). Alternatively, stained cells were washed with probe-free medium and the samples were viewed using fluorescence microscopy (Olympus BX 40).

### 2.10. Statistical Analysis

Data are reported as the means ± SDs. Statistical analyses were performed using SigmaPlot 11.0 software package (Systat Software Inc., San Jose, CA, USA). The statistical significance of differences was determined by Student’s *t*-test (when the population means of only two groups were to be compared), and one-way ANOVA (when means of more than two groups were to be compared). We used * or # for the significant results (*p* < 0.05) and ** or ## for the very significant results (*p* < 0.01).

## 3. Results

### 3.1. Lysosomal Sequestration of TKIs

Here we addressed the question of what amount of TKIs are sequestered in lysosomes and whether this mechanism itself is sufficient to mediate significant resistance in cancer cells in vitro. In general, TKIs belong to the broad group of hydrophobic, weak-base drug chemotherapy agents, which may accumulate in lysosomes [6]. We studied three TKIs, IM, NIL, and DAS which are currently used for the treatment of patients with diagnoses of chronic myeloid leukemia [30]. Molecular structures and basic physicochemical properties of IM, NIL, and DAS are given in Table 1.

First, we determined absolute and relative accumulation of TKIs in lysosomes of cancer cells. As expected, we found that IM, NIL, and DAS were significantly accumulated in the lysosomes of cancer cells (Figure 2, Table 1). Of the TKIs studied, IM was the most sequestrated in lysosomes of cancer cells (Figure 2a, Table 1). The absolute lysosomal accumulation of TKIs increased with increasing extracellular concentration without reaching a plateau (Figure 2a,c,e). Interestingly, relative accumulation of TKIs in lysosomes was also concentration dependent: the higher the extracellular TKI concentration, the greater was the percentage of drug accumulated in lysosomes (Figure 2b,d,f). Here too, the effect was most pronounced for IM (Figure 2b, Table 1).

### 3.2. Lysosomal Sequestration of TKIs and Extracellular Drug Level

Next, we calculated how much the lysosomal sequestration of TKIs can affect the extracellular drug levels, since direct measurement was impossible. Therefore, we compared the total amount of a particular TKI added to the growth medium (100%) with the absolute amount of this TKI accumulated in lysosomes of cancer cells. Using this approach, we found that the amount of drug accumulated in lysosomes represented a minor fraction of the amount of drug added to the growth medium (Figure 3). In accordance with these results, we observed that lysosomal sequestration of TKIs did not change their extracellular levels dramatically (Figure 3). The only exception was IM; however, its accumulation was ≤10% of the total amount of IM in growth medium (Figure 3a). These results clearly showed that lysosomal sequestration of TKIs did not substantially affect their extracellular concentrations and, therefore, neither their extralysosomal concentrations (target sites).

### 3.3. Lysosomal Sequestration of TKIs and Their Accessibility to Intracellular Targets

We further asked whether lysosomal sequestration of TKIs could affect the sensitivity of cellular targets. According to the currently accepted hypothesis, lysosomal drug sequestration contributes to drug resistance [5,20]. We could then expect a difference in inhibition efficacy between cells with abrogated lysosomal sequestration of TKIs and unaffected cells. However, we observed that lysosomal accumulation of IM, NIL, and DAS failed to affect the Bcr-Abl signaling, as judged by the pCrkL (Tyr 207) and pBcr (Tyr 177) dephosphorylation (Figure 4). These results collectively indicated that even significant lysosomal drug accumulation can hardly result in drug resistance.

### 3.4. Stimulation of Lysosomal Sequestration Capacity by Sunitinib

The short term (72 h) incubation of cells with sub-cytotoxic concentrations of hydrophobic weak-base chemotherapeutics induces lysosomal biogenesis in cancer cells [20]. For this reason, we incubated cells with low concentrations of SUN to enhance the sequestration capacity of their lysosomes. As expected, SUN treated cells exhibited increased lysosomal sequestration capacities, as judged by the increased staining using lysotracker red (Figure 5). Accordingly, SUN-pretreated cells exhibited increased lysosomal accumulation capacity for TKIs, including IM, NIL, and DAS (Figure 6). However, the increased lysosomal sequestration capacity was not due to the lysosomal enlargement, since lysosomal proteins, including LAMP1, LAMP2, vacuolar ATPase subunit B2, ACP, and GLB did not exhibit increased expression or activity (Figure 7).

### 3.5. Sequestration of TKIs in Sunitinib-Stimulated Lysosomes and Extracellular Drug Levels

However, even enhanced lysosomal sequestration of IM, NIL, and DAS induced by 72 h pre-incubation with SUN failed to reduce their extracellular concentrations dramatically for clinically relevant drug concentrations (Figure 8). Accordingly, the amount of IM, NIL, or DAS sequestrated in lysosomes represented a minor fraction of the total amount of the particular TKI added to the growth medium (Figure 8). The only exception was the highest IM concentration (30 µM) that decreased extracellular IM levels dramatically (Figure 8). However, such concentrations are clinically irrelevant [31].

### 3.6. The Sequestration of TKIs in Sunitinib-Stimulated Lysosomes and Their Accessibility to Intracellular Targets

We measured whether increased lysosomal sequestration of TKIs induced by SUN could affect the sensitivity of cells to these drugs. Our results clearly indicated that despite increased lysosomal accumulation of TKIs in SUN-stimulated cells, the Bcr-Abl inhibition was unaffected by BafA1. In other words, even enhanced lysosomal accumulation of TKIs did not change the cellular sensitivity to these drugs (Figure 9). Here too, the results indicated that even enhanced lysosomal accumulation of the TKIs did not result in drug resistance.

## 4. Discussion

Lysosomal drug sequestration was theoretically explained many years ago [32,33]. The theory required the assumption that hydrophobic, weak-base drugs cannot diffuse across membranes in their ionized (charged) state, whereas in their non-ionized state, they can do so freely. Since lysosomes have low internal pHs (≤5) and the pH in the cytosol is close to neutral, lysosomes accumulate large amounts of these drugs [32,33]. Later on, this phenomenon was associated with the emergence of multidrug resistance [1,5]. In this context, it was assumed that lysosomal sequestration simultaneously reduces drug availability in the target-containing compartment, and thus reduces its therapeutic efficacy [5]. Recently, a similar idea was elaborated by Zhitomirsky and Assaraf, who postulated that the “lysosomal sequestration of hydrophobic, weak-base anticancer drugs prevents their accessibility to their intracellular target sites” [20]. In other words, lysosomal sequestration decreases the extra-lysosomal drug concentration (drug concentration in the intracellular target sites).

If the above behavior of hydrophobic weak-base anticancer drugs is real in cells, it should be possible to measure it. However, direct assessment of cell sensitivity to TKIs in the presence and absence of inhibitors of lysosomal sequestration such as BafA1 and NH_4_Cl was not possible here, since both compounds were cytotoxic themselves for all cell lines used in our experiments (not shown). For this reason, we had to rely on indirect methods.

Hence, we attempted a novel approach which enabled us to measure whether lysosomal sequestration of TKIs actually reduces their concentration in the extralysosomal space (at the target sites; i.e., cytosol). We reasoned that a lower concentration of drug in the extralysosomal space could be monitored by the measurement of extracellular drug concentration. Indeed, only lysosomal accumulation to the extent that it measurably decreased the extracellular drug concentration significantly reduced the extralysosomal drug concentration, and thus affected the sensitivity to the drug in vitro (Figure 1c). This is due to the equilibrium between extracellular, intracellular (extralysosomal), and intralysosomal spaces (Figure 1c), as uncharged forms of weakly basic drugs can freely diffuse across cellular membranes [5,20]. In other words, the lysosomal sequestration of weak-base drugs cannot reduce their extralysosomal (target sites) concentrations without affecting the extracellular drug concentrations because of the equilibrium among lysosomes, the extralysosomal space, and the extracellular space (Figure 1c). Since the extracellular space is incomparably greater than the lysosomal space within in vitro experiments, the sequestration of weakly basic drugs by lysosomes would have to be enormous to significantly alter their extracellular concentrations.

In the present work, we analyzed how much lysosomal sequestration of IM, NIL, and DAS could affect the extracellular drug concentration, and thus, the cell sensitivity to these drugs. We observed that all TKIs we studied, including IM, NIL, and DAS, were sequestrated into the lysosomes of cancer cells (Figure 2). Importantly, this process seemed to depend on the TKI concentration: the higher the extracellular TKI concentration, the greater was its absolute and relative accumulation in lysosomes (Figure 2). Of those drugs, IM accumulated the most (Figure 2). This is not surprising considering the physicochemical properties of TKIs we used (Table 1). However, lysosomal sequestration affected the extracellular TKI concentrations only marginally (Figure 3). Even the highest accumulation of IM was only able to induce less than a 10% decrease in its extracellular level (Figure 3). This could hardly compromise IM efficiency. In the cell lines we used, A549, K562, and MCF7, there were no large differences in the lysosomal accumulations of TKIs (Figure 2 and Figure 3). Probably, the weakest ability to accumulate TKIs was exhibited by the MCF7 cell line (Figure 2 and Figure 6).

The three-day incubation of cancer cells with SUN-enhanced the sequestration capacity of their lysosomes for a fluorescent probe (Figure 5) and all of the TKIs studied (Figure 6). In contrast to others [20], we observed that this effect was not due to lysosomal biogenesis, since the expression of lysosomal proteins, including LAMP1, LAMP2, and vacuolar ATPase subunit B2, was unaffected (Figure 7). The enzyme activity of ACP and GLB also remained obviously unchanged (Figure 7). We can speculate that the increased lysosomal capacity was due to other mechanisms, such as decreased lysosomal pH or changed ion distribution. However, this issue must be further studied.

Similarly, even in cells with SUN-enhanced sequestration capacity, the extracellular levels of studied TKIs also decreased only by a few percent (≤10%; Figure 8). The only exception was IM, where lysosomal sequestration of 30 µM IM dramatically (≥10%) reduced its extracellular concentration in all cell lines studied, and thus, could theoretically mediate resistance (Figure 8). Importantly, this effect only occurred for clinically irrelevant IM concentrations [31].

Another indirect method for verifying whether lysosomal sequestration of drugs reduces their concentration in the extralysosomal space, is to assess the sensitivity of target molecule(s) to these drugs. In other words, inhibition of lysosomal drug sequestration should increase sensitivity of target molecule(s) to this drug. In our experimental setup, we measured the sensitivity of Bcr-Abl signaling to the TKIs in the presence and/or absence of BafA1 in K562 cells. Analysis of the Bcr-Abl signaling clearly indicated that lysosomal sequestration itself failed to affect inhibition efficacy for any of the TKIs, as judged by the phosphorylation pattern of Bcr-Abl (Tyr 177) and CrkL (Tyr 207; Figure 4), the critical leukemogenic proteins [34,35]. Similarly, even increased lysosomal accumulation of TKIs in SUN-stimulated cells did not change the sensitivity of Bcr-Abl signaling pathway to the drugs (Figure 9). Inhibition of TKI sequestration by BafA1 or NH_4_Cl (not shown) had no effect on the phosphorylation pattern of Bcr-Abl (Tyr 177) or CrkL (Tyr 207; Figure 9). These results are in good agreement with the findings of other researchers, who reported that lysosomal sequestration of IM or SUN had no effect on oncogenic signaling [15,19]. Our experimental design that monitored Bcr-Abl signaling also allowed us to explore other possibilities. First, Bcr-Abl signaling may also serve as a free drug biomarker. This is important since added drug concentration may also be sensitive to other depletion, such as non-specific binding in the assay. Second, the role of lysosomes as part of cell signaling is now recognized [36]. In this context, lysosomal drug accumulation could trigger possible changes in drug-target states remote from the lysosome. This effect, unlike simple depletion, could have in vivo consequences and must be studied further.

## 5. Conclusions

Lysosomal sequestration of an anticancer drug can significantly decrease its concentration in extralysosomal space (at target sites) only when it simultaneously decreases its extracellular concentration due to equilibrium, since the uncharged form of a weak-base drug freely diffuses across cellular membranes. Even though IM, NIL, and DAS were extensively accumulated in the lysosomes of cancer cells, their sequestration was not sufficient to substantially reduce the extracellular drug concentration. In accordance with these results, the lysosomal accumulation of TKIs did not affect the Bcr-Abl tyrosine kinase signaling. The three-day treatment of cells with a low concentration of SUN significantly enhanced lysosomal sequestration capacity; however, without apparent lysosomal biogenesis. Importantly, even SUN-enhanced lysosomal sequestration of IM, NIL, and DAS did not change their availability in the target-containing compartment, and thus, can hardly contribute to any drug resistance in in vitro experiments.

## Figures and Tables

**Figure 1 biomolecules-09-00675-f001:**
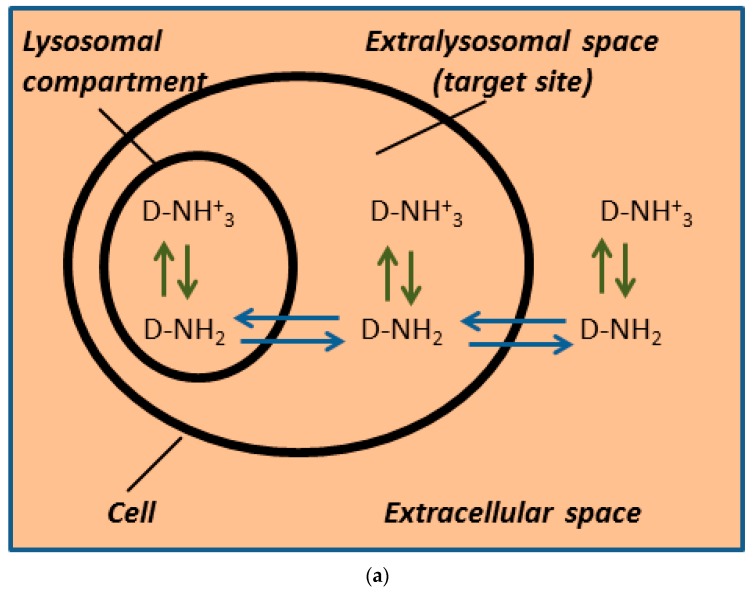
Weak-base drug distribution within the cell in vitro. Our model includes several theoretical assumptions: (i) uncharged forms of weakly basic drugs can freely diffuse across cellular membranes [5,20]; (ii) only two interactions are considered, the Henderson–Hasselbach equilibrium ↑ ↓ and the passive diffusion of uncharged molecules ↔; (iii) the extra and intracellular (extralysosomal space) pHs are equal. Color saturation corresponds to the drug concentration. (**a**) Drug distribution without lysosomal sequestration. (This may occur, for example, after the addition of BafA1, a vacuolar ATPase inhibitor.) Uncharged molecules can freely diffuse across cell membranes. (**b**) Drug distribution with lysosomal sequestration. Uncharged molecules can freely diffuse only across the lysosomal membrane. Under such conditions, the lysosomal sequestration of a drug can dramatically reduce its extralysosmal concentration (drug concentration at target sites), and thus mediate drug resistance. However, such a model is not applicable since it does not fulfil theoretical assumptions. In fact, the diffusion of uncharged molecules across the plasma membrane would dissipate the gradient between extra and intracellular space. (**c**) Drug distribution with lysosomal sequestration. Uncharged molecules can freely diffuse across cell membranes. The lysosomal sequestration of a drug that reduces drug concentration at target sites (= in extralysosomal space) and simultaneously decreases the extracellular drug level can mediate resistance to this drug. This is the only model to fulfil theoretical assumptions.

**Figure 2 biomolecules-09-00675-f002:**
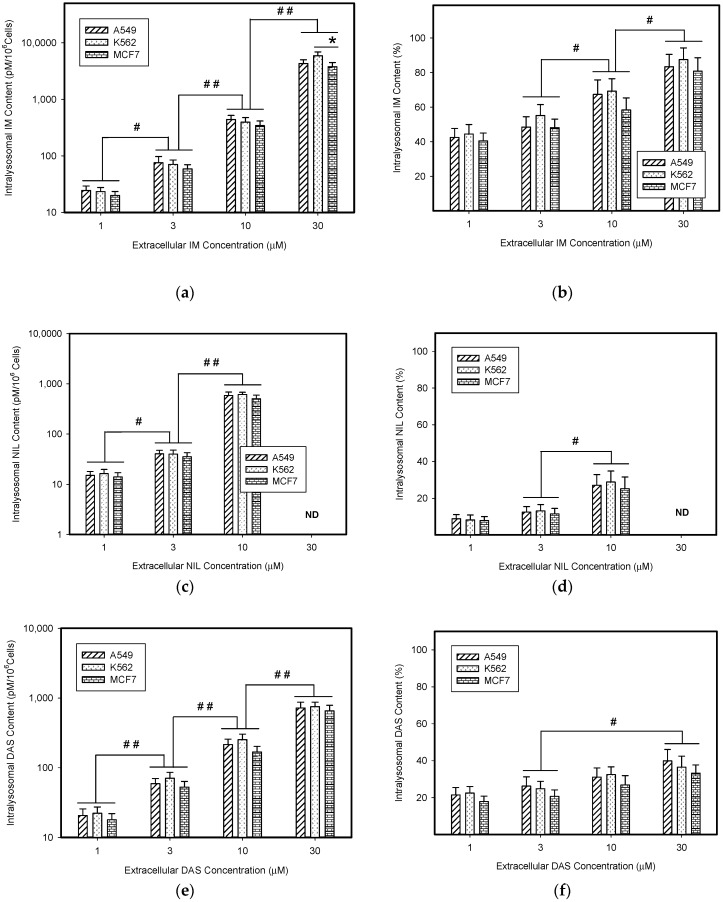
Lysosomal sequestration of TKIs. (**a**) Absolute accumulation of IM in lysosomes of cancer cells. (**b**) Relative accumulation of IM in lysosomes of cancer cells. (**c**) Absolute accumulation of NIL in lysosomes of cancer cells. (**d**) Relative accumulation of NIL in lysosomes of cancer cells. ND—not determined (due to limited NIL solubility). (**e**) Absolute accumulation of DAS in lysosomes of cancer cells. (**f**) Relative accumulation of DAS in lysosomes of cancer cells. Columns represent the means and standard deviations of four independent experiments. * denotes a significant change in the intralysosomal IM content (*p* < 0.05) between the indicated cell lines. # denotes significant change in the intralysosomal content of the appropriate TKI (*p* < 0.05) between the groups indicated. ## denotes a very significant change in the intralysosomal content of the appropriate TKI (*p* < 0.01) between the groups indicated.

**Figure 3 biomolecules-09-00675-f003:**
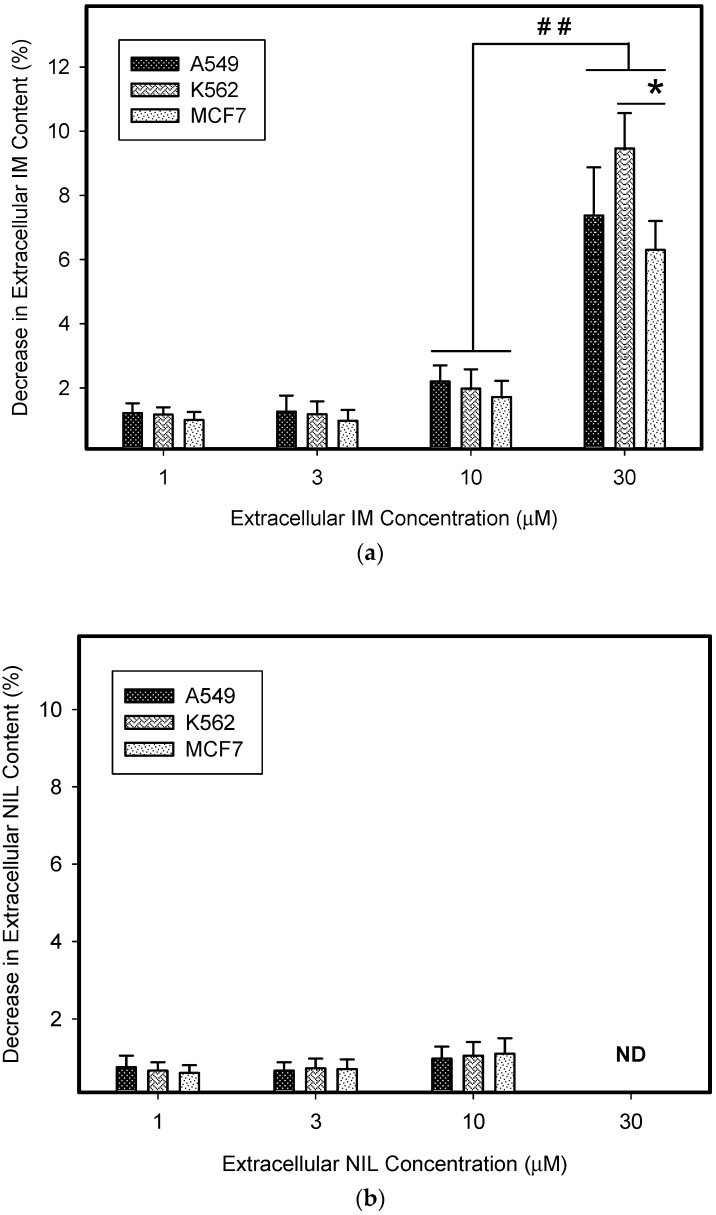
Lysosomal sequestration of TKIs and extracellular drug concentration. (**a**) The effect of lysosomal sequestration of IM on its extracellular concentration. (**b**) The effect of lysosomal sequestration of NIL on its extracellular concentration. ND—not determined (due to limited NIL solubility). (**c**) The effect of lysosomal sequestration of DAS on its extracellular concentration. Columns represent the means and standard deviations of four independent experiments. * denotes a significant change in the decrease in extracellular IM content (*p* < 0.05) between the indicated cell lines. ## denotes a very significant change in the decrease in extracellular IM content (*p* < 0.01) between the indicated groups.

**Figure 4 biomolecules-09-00675-f004:**
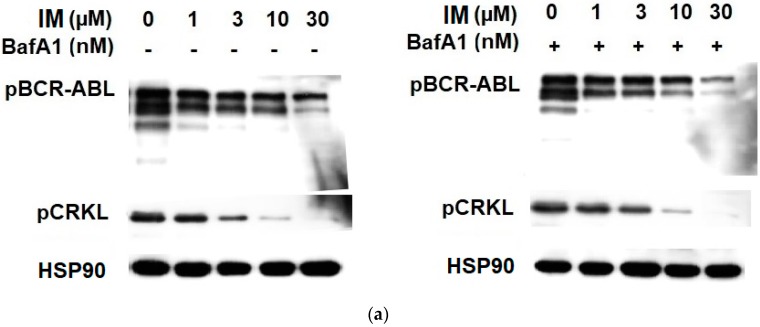
The effect of the lysosomal sequestration of TKIs on the inhibition of Bcr-Abl signaling. Cells were incubated with various TKI concentrations in the absence or presence BafA1. After 6 h, cell extracts were analyzed for CrkL phosphorylation (Tyr207) and Bcr-Abl phosphorylation (Bcr (Tyr177)) using western blot analysis. (**a**) Cells incubated wit IM ± BafA1 (typical analysis). (**b**) Quantitative analysis of western blots using densitometry. (**c**) Cells incubated wit NIL ± BafA1 (typical analysis). (**d**) Quantitative analysis of western blots using densitometry. (**e**) Cells incubated wit DAS ± BafA1 (typical analysis). (**f**) Quantitative analysis of western blots using densitometry. Columns represent the means and standard deviations of four independent experiments.

**Figure 5 biomolecules-09-00675-f005:**
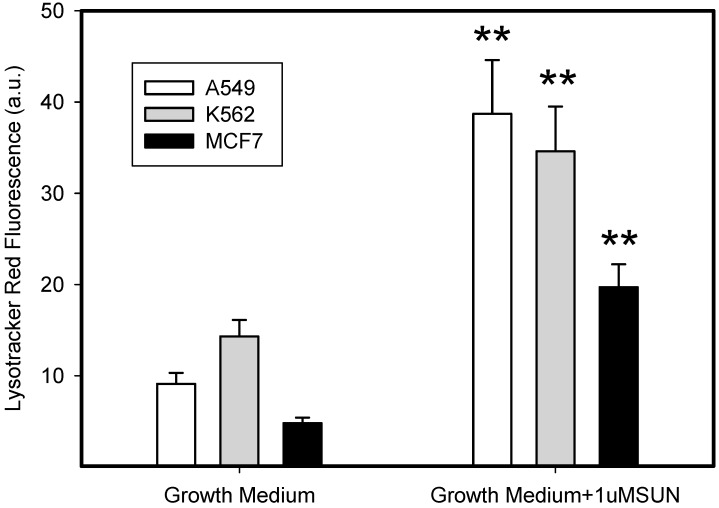
The effect of SUN on the lysosomal sequestration capacity of cancer cells. Cells were cultured for 3 days in the presence of 1 µM SUN under standard conditions. Cells cultured in medium without SUN were taken as controls. The lysosomal sequestration capacity of cells was analyzed using flow cytometry upon staining with Lysotracker Red. Columns represent the means and standard deviations of four independent experiments. ** denotes a significant change in the mean fluorescence (*p* < 0.01) between respective SUN-stimulated and SUN-unstimulated cells.

**Figure 6 biomolecules-09-00675-f006:**
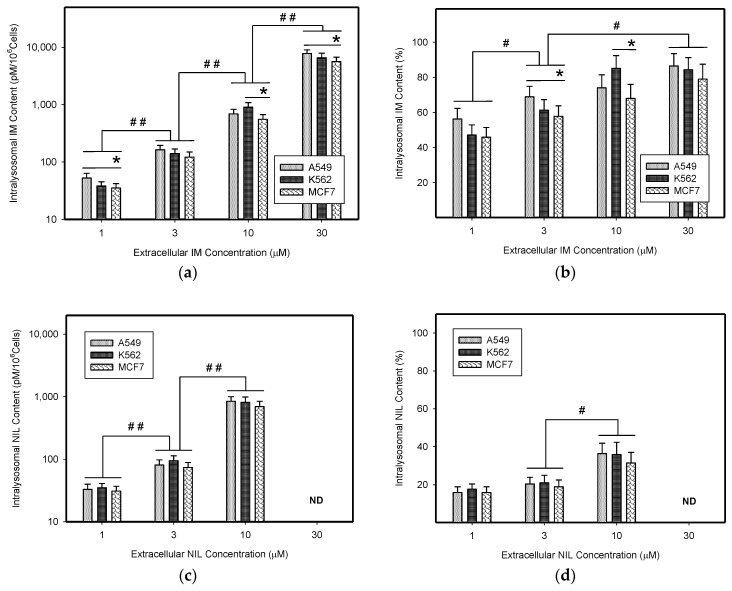
Accumulation of TKIs in the lysosomes of SUN-stimulated cancer cells. Cells were cultured for 3 days in the presence of 1 µM SUN under standard conditions. Cells cultured in medium without SUN were taken as controls. (**a**) Accumulation of IM (absolute value). (**b**) Accumulation of IM (relative value). (**c**) Accumulation of NIL (absolute value). ND—not determined (due to limited NIL solubility). (**d**) Accumulation of NIL (relative value). (**e**) Accumulation of DAS (absolute value). (**f**) Accumulation of DAS (relative value). Columns represent the means and standard deviations of four independent experiments. * denotes a significant change in the intralysosomal IM content (*p* < 0.05) between the indicated cell lines. # denotes a significant change in the intralysosomal content of appropriate TKI (*p* < 0.05) between the groups indicated. ## denotes very significant change in the intralysosomal content of appropriate TKI (*p* < 0.01) between the groups indicated.

**Figure 7 biomolecules-09-00675-f007:**
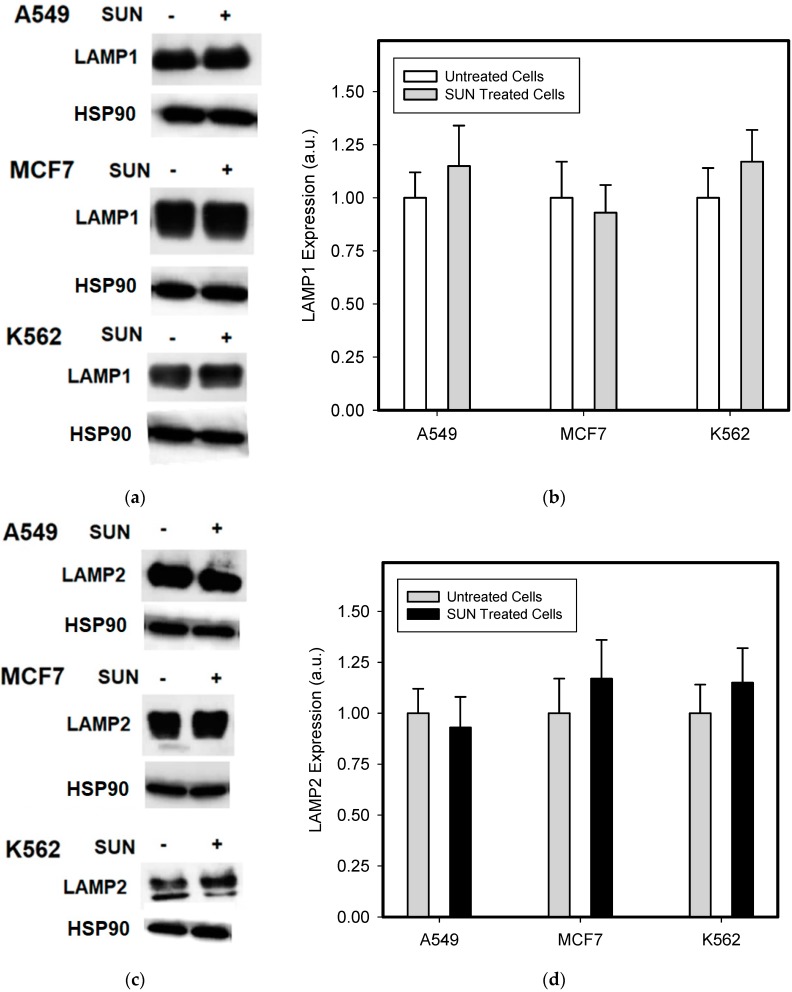
The effect of SUN on the expression of lysosomal proteins in cancer cells. Cells were cultured for 3 days in the presence of 1 µM SUN under standard conditions. Cells cultured in medium without SUN were taken as controls. (**a**) Western blot analysis of LAMP1 (typical analysis). (**b**) Quantitative analysis of LAMP1 expression using densitometry. (**c**) Western blot analysis of LAMP2 (typical analysis). (**d**) Quantitative analysis of LAMP2 expression using densitometry. (**e**) Western blot analysis of vacuolar ATPase subunit B2 (typical analysis). (**f**) Quantitative analysis of vacuolar ATPase subunit B2 using densitometry. (**g**) Enzymatic activity of lysosomal APC. (**h**) Enzymatic activity of lysosomal GLB. Columns represent the means and standard deviations of four independent experiments.

**Figure 8 biomolecules-09-00675-f008:**
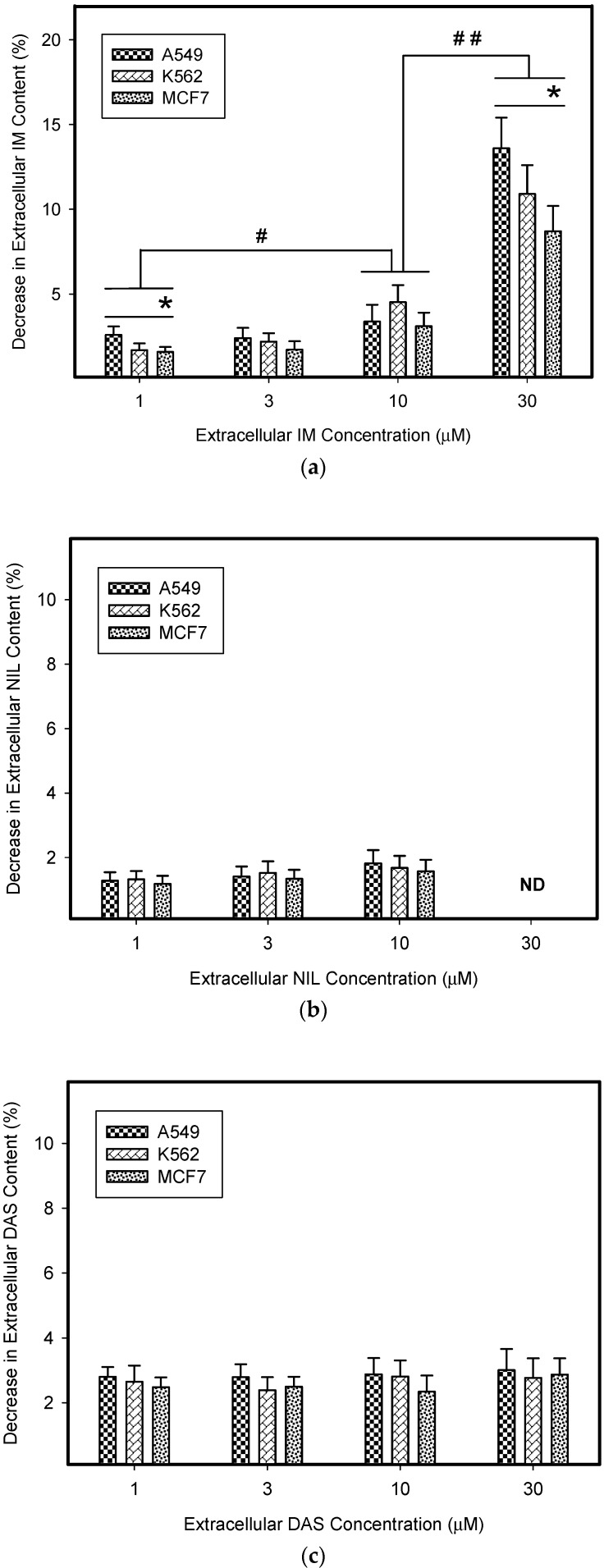
Lysosomal sequestration of TKIs in SUN-stimulated cells and extracellular drug concentrations. (**a**) The effect of the lysosomal sequestration of IM in SUN-stimulated cells on its extracellular concentration. (**b**) The effect of the lysosomal sequestration of NIL in SUN-stimulated cells on its extracellular concentration. ND—not determined (due to limited NIL solubility). (**c**) The effect of the lysosomal sequestration of DAS in SUN-stimulated cells on its extracellular concentration. Columns represent the means and standard deviations of four independent experiments. * denotes a significant change in the decrease in extracellular IM content (*p* < 0.05) between the cell lines indicated. # denotes significant change in the decrease in extracellular IM content (*p* < 0.05) between the groups indicated. ## denotes very significant change in the decrease in extracellular IM content (*p* < 0.01) between the groups indicated.

**Figure 9 biomolecules-09-00675-f009:**
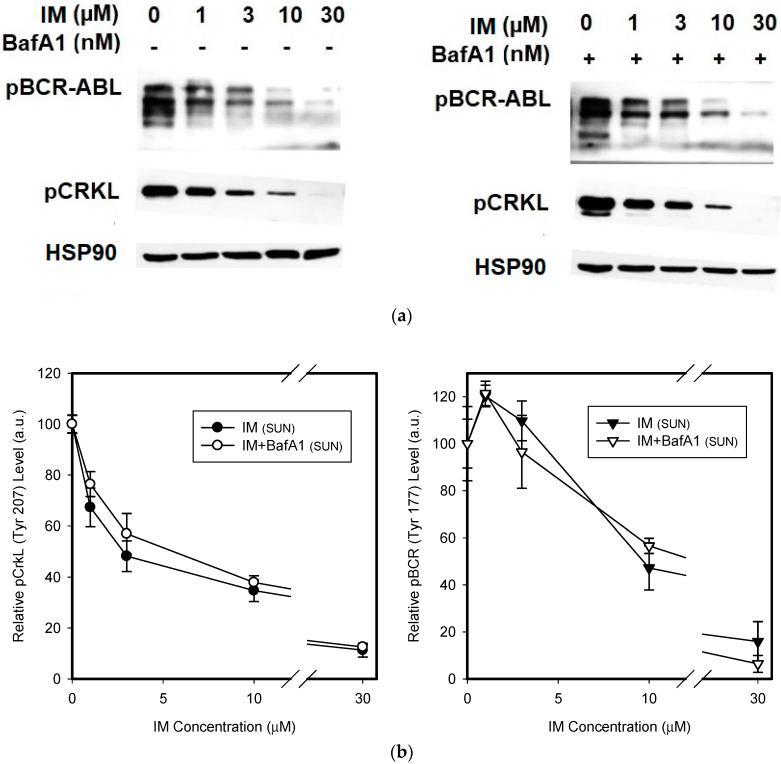
The effect of the lysosomal sequestration of IM on Bcr-Abl signaling in SUN-stimulated cancer cells. SUN-stimulated cells were incubated with various TKIs concentration in the absence or presence BafA1. After 6 h, cell extracts were analyzed for the CrkL phosphorylation (Tyr207) and Bcr-Abl phosphorylation (Bcr (Tyr177)) using western blot analysis. (**a**) Cells incubated wit IM ± BafA1 (typical analysis). (**b**) Quantitative analysis of western blots using densitometry. (**c**) Cells incubated wit NIL ± BafA1 (typical analysis). (**d**) Quantitative analysis of western blots using densitometry. (**e**) Cells incubated wit DAS ± BafA1 (typical analysis). (**f**) Quantitative analysis of western blots using densitometry. Columns represent the means and standard deviations of four independent experiments.

**Table 1 biomolecules-09-00675-t001:** Molecular structure and physicochemical properties of the TKIs studied.

Drug	pKa ^1^	log P ^2^	Molecular Structure ^3^
Imatinib	8.27	4.38	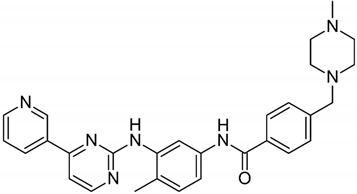
Nilotinib	5.92	5.36	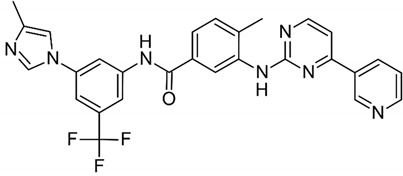
Dasatinib	7.22	3.82	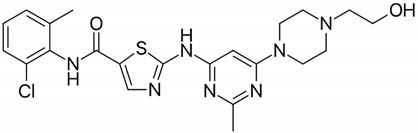

^1^ pKa values referring to the strongest basic residue in each molecule were predicted using ChemAxon software and were adapted from DrugBank. ^2^ logP values were predicted using ChemAxon software and were adapted from DrugBank. ^3^ Molecular structures were adapted from DrugBank.

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
