# Peer review of "The Lysosomal Sequestration of Tyrosine Kinase Inhibitors and Drug Resistance"

_biomolecules, 2019, doi:10.3390/biom9110675_

Round 1
Reviewer 1 Report
This is a very important topic. The studies are nicely designed and executed. The manuscript is very well written and the conclusions are clear. The findings are very significant. In Figure 1, a, b and c all looked the same. It might be good to add some “molecules” to illustrate the different models.
Author Response
Reviewer #1:
We thank the Reviewer for careful reading of our MS. We are delighted that you consider our article is well-written with clearly worded conclusions and that you evaluate our results as valuable. We have modified it according to your suggestions. Changes are in red.
Your Comment:
This is a very important topic. The studies are nicely designed and executed.
The manuscript is very well written and the conclusions are clear.
The findings are very significant.
Response:
Thank you!
In Figure 1, a, b and c all looked the same. It might be good to add some “molecules” to illustrate the different models.
Response:
Unfortunately, we are unable to add results with other molecules to demonstrate different properties. First, we also studied gefitinib or daunorubicin but their ability to accumulate in lysosomes of cancer cells was very similar to those of IM, NIL, and DAS. Second, the number of results (figures) within such a MS would be ponderous. Instead we added a table with the molecular structures of studied molecules together with their physicochemical properties. Please, see Table 1 in the revised version of the MS.
Reviewer 2 Report
The current manuscript of Ruzickova et al. is largely based on the hypothesis of Figure 1, that was mostly not demonstrated. In addition Figure 1 is not a result and therefore should be moved to the introduction and mostly discussed in the discussion section. Moreover the experimental section lacks important and crucial details and experiments: did the authors perform analysis in triplicates or more? please detail how. Also most figures are lacking appropriate statistical analysis, for examples, see figures 2 and 3. What the star stand for in figure 5? In the current form and missing all these experiments and details, the manuscript is far too preliminary to deserve to be reported.
Author Response
Reviewer #2:
We thank the Reviewer for careful reading of our MS. According to your suggestions, we have made substantial changes and adjustments. We hope that the revised version of our MS is now acceptable for publication in the Biomolecules. Changes are in green.
Your Comment:
1. The current manuscript of Ruzickova et al. is largely based on the hypothesis of Figure 1, that was mostly not demonstrated. In addition Figure 1 is not a result and therefore should be moved to the introduction and mostly discussed in the discussion section.
Response:
The MS has been modified accordingly. Figure 1 is moved to the Introduction. Please, see revised version of the MS.
2. Moreover, the experimental section lacks important and crucial details and experiments: did the authors perform analysis in triplicates or more? please detail how.
Response:
At this point we cannot agree with you. All experiments were done as four independent repetitions. This information is provided in each figure legend of the original MS version. It is now highlighted in the revised version MS.
3. Also most figures are lacking appropriate statistical analysis, for examples, see figures 2 and 3.
Response:
Statistical data processing has been rewritten in order to make it more clear. Please, see section Materials and Methods (2.10. Statistical Analysis). According to your suggestion, statistical analysis is provided for all graphs when needed.
4. What the star stand for in figure 5?
Response:
The statistical analysis was performed again. The explanation is given in the text below the graph.
5. In the current form and missing all these experiments and details, the manuscript is far too preliminary to deserve to be reported.
Response:
On this point we cannot agree with you. First, all assay methods are described in Materials and Methods in details. Second, our study was done with three cancer cell lines A549, K562, and MCF7, and for three drugs, IM, NIL, and DAS. The experiments were done in four independent repetitions. The overall work on this MS took almost two years. We do not consider the results preliminary.
Reviewer 3 Report
The presented paper is well written and could be published in its current form.
Author Response
Reviewer #3:
We thank the Reviewer for careful reading of our MS. We are delighted that you consider our article to be well-written. We are pleased that you suggest the MS could be published in its current form.
Round 2
Reviewer 2 Report
Most of comments from the previous version have not been corrected.
Major points:
1) The current manuscript of Ruzickova et al. is largely based on the hypothesis of Figure 1, that was mostly not demonstrated.
2) statistical analysis have been performed between cell lines but not between the different concentration. Thus dose-dependence is not assessed.
In the current form and missing all these experiments and details, the manuscript is far too preliminary to deserve to be reported.
Author Response
We thank the Reviewer for reading of our MS. According to your suggestions, we have made additional changes and adjustments. We hope that the revised version of our MS is now acceptable for publication in Biomolecules. Changes are in cyan.
Major points:
Your Comment:
1) The current manuscript of Ruzickova et al. is largely based on the hypothesis of Figure 1, that was mostly not demonstrated.
Response: Unfortunately, we can not agree with you at this point. Please, read once more our MS.
2) statistical analysis have been performed between cell lines but not between the different concentration. Thus dose-dependence is not assessed.
Response:
Statistical analysis was extended according to your suggestion, please see revised MS.
Round 3
Reviewer 2 Report
The corrections have been poorly (or not) performed since the 1st review. I still do not agree with the discussion on the hypothesis. The dose-response is not discussed.